# A feature-based approach for atlas selection in automatic pelvic segmentation

**Guoping Shan**[1,2], **Xue Bai**[2], **Yun Ge**[1]*, **Binbing Wang** [2]*

**1** School of Electronic Science and Engineering, Nanjing University, Nanjing, Jiangsu, China,
**2** Department of Radiation Physics, Zhejiang Key Laboratory of Radiation Oncology, Zhejiang Cancer Hospital, Hangzhou Institute of Medicine (HIM), Chinese Academy of Sciences, Hangzhou, Zhejiang, China

* geyun@nju.edu.cn (YG); wangbb@zjcc.org.cn (BBW)

## Abstract

Accurate and efficient automatic segmentation is essential for various clinical tasks such as radiotherapy treatment planning. However, atlas-based segmentation still faces challenges due to the lack of representative atlas dataset and the computational limitations of deformation algorithms. In this work, we have proposed an atlas selection procedure (subset atlas grouping approach, MAS-SAGA) which utilized both image similarity and volume features for selecting the best-fitting atlases for contour propagation. A dataset of anonymized female pelvic Computed Tomography (CT) images demonstrated that MAS-SAGA significantly outperforms conventional multi-atlas-based segmentation (cMAS) in terms of Dice Similarity Coefficient (DSC) and 95th Percentile Hausdorff Distance (95HD) for bladder and rectum segmentation using a three-fold cross-validation strategy. The proposed procedure also reduced computation time compared to cMAS, making it a promising tool for medical image analysis applications. In addition, we have evaluated two distinct atlas selection methods: the Feature-based Atlas Selection Approach (MAS-FASA) and the Similarity-based Atlas Selection Approach (MAS-SIM). We investigate the differences between these two methods in terms of their ability to select the best fitting atlases. The findings demonstrated that MAS-FASA selected different atlases than MAS-SIM, resulting in improved segmentation performance overall. It highlighted the potential of feature-based subgrouping techniques in enhancing the efficacy of MAS algorithms in the field of medical image segmentation.

## Introduction

Medical image segmentation predefines normal tissues for the purpose of their protection in radiation therapy planning, thus having broad applications in the field of radiation therapy [1,2]. Multi-atlas-based segmentation (MAS) uses the prior knowledge provided by finding the best-fitting images contoured previously and then propagating the delineation in the atlas onto the target image using Deformable Image Registration (DIR) [3]. In addition to image segmentation, MAS is also an important method for radiation therapy dose accumulation and prediction [4,5,6]. Unlike Deep Learning-based methods that typically predict the likelihood of each pixel belonging to a certain class, MAS calculates Deformation Vector Fields (DVF),

**Data availability statement:** "Data cannot be shared publicly because it contains personal information restricted to use. Data are available from the Zhejiang Cancer Hospital Institutional Data Access / Ethics Committee contact via (ec@zjcc.org.cn) for researchers who meet the criteria for access to confidential data."

**Funding:** This study was financially supported by the Medical Science and Technology Project of Zhejiang Province (2021PY039), the Natural Science Foundation of Zhejiang Province (LSY19H180002). The funders had no role in study design, data collection and analysis, decision to publish, or preparation of the manuscript.

which provide an elastic mapping between image coordinates to adjust the alignment between the atlases and the target image. The DVF provided by such type of registration approach is continuous and interpretable. Compared to Deep Learning-based approaches which require extensive amounts of labeled data or computational resources, MAS offers a practical solution that only requires finite training data and yields accurate results across various imaging modalities.

MAS is typically composed of three steps: atlas selection, registration, and label fusion [7,8]. The choice of atlas selection significantly impacts the accuracy of image segmentation. Intensity-based similarity metric is a widely used method to find most fitting atlases. These similarity metric includes similarity index [9], the sum of squared difference of image intensity [10], correlation coefficient [11], and mutual information (MI) [12]. Although these methods can quantify the overall similarity of the entire image, global similarity measures may be insensitive to local variations within specific anatomical structures, leading to less accurate comparisons in regions with unique imaging.

Recently, MAS has been challenged by deep learning-based segmentation. Deep learning based segmentation employs deep neural network models to learn features and semantic information from images, achieving excellent segmentation results in medical image segmentation [13]. However, this method also exhibits certain drawbacks, such as challenges in data acquisition and annotation, limited model generalization capability, and poor interpretability. When employed in new tasks or with diverse types of imaging data, their performance may significantly decline [14]. In contrast, MAS is an interpretable method and does not require a large number of annotated images for training. Therefore, it is still being used in some fields, such as brain segmentation [15] and dose accumulation assessment in radiotherapy [16,17] and has not been replaced by deep learning. however, the lack of precise feature classification during the atlas search and deformation processes in MAS leads to a segmentation accuracy inferior to that achieved by deep learning methodologies.

Some efforts have been made in determining the most representative atlas groups by combining image similarity with other image features. For instance, previous study using Location-Based Feature Matching atlas pre-selection approaches and compared them to random and Mutual Information-based Methods [18] and genetic algorithm [19]. Zaffino et al. introduced an approach for selecting preregistration atlas subsets. They based this method on the pairwise feature selection of both the whole prostate and the left ventricle of the heart from magnetic resonance images. This approach relies on selecting the best performing group of atlases rather than the group of highest scoring individual atlases [20]. Another study reported an iterative atlas selection procedure with a cross-validation strategy where each dataset serves as an atlas set to segment each image in the other dataset [21]. Other approaches such as the selection of atlases using manifold parameters in head and neck CT images [22] and neighborhood approximation forests [23] have also been proposed in the literature. However, these methods are often task-specific for particular segmentation and the extraction of key features as well as the specific deformation algorithms used subsequently can significantly influence the segmentation performance.

Our study is based on the following hypothesis that similarity-based atlas selection methods tend to search for atlases with high overall image similarity. However, when dealing with large-scale deformation, the accuracy of deformation modeling is often compromised due to the involvement of significant nonlinear deformations, geometric complexities, and the impact of intricate boundary conditions and constraints. It may lead to instability in segmentation results, making further refinement challenging. To assess the improvement of segmentation performance using different atlas selection methods, we conducted experiments comparing them against a conventional multi-atlas segmentation (cMAS) approach. We firstly

propose an atlas grouping method (subset atlas grouping approach, SAGA). This method utilizes both image similarity and volume features to enhance the accuracy of the Deformable Image Registration (DIR) step. By extracting image features and performing classification, subsets corresponding to specific classification tasks are established. We also suggested a feature-based atlas selection approach (MAS-FASA), where the atlas with the closest feature space distance is selected as the candidate for MAS segmentation. This method eliminates the need for additional atlases, resulting in reduced atlas search time.

Specifically, the contributions of our work are as follows:

1. A subgrouping atlas search approach was proposed. This enables the search strategies to select the most fitting atlases considering both similarity and volume features, thereby enhancing segmentation accuracy.

2. To further clarify the advantage of volume features in selecting atlases, this study then ranked the most fitting atlases obtained from two atlas selection approaches, based on similarity and volume features, and compared their differences in priority when selecting candidate atlases.

3. A comparison of the execution time efficiency of the four proposed atlas search methods was also performed.

Finally, we compared the differences in atlases obtained through similarity-based selection and feature-based selection methods to verify significant difference in segmentation performance. Moreover, we analyzed the different images selected by these methods and the computational time required for each approach.

## Materials and methods

### Conventional Multi-Atlas Segmentation (cMAS)

Conventional Multi-Atlas Segmentation (cMAS) is a widely used technique in medical imaging applications that utilizes prior knowledge provided by contoured atlas images to perform segmentation tasks on un-contoured target images. The process begins by selecting the best-fitting contoured atlas images, followed by identifying the corresponding DVF between the target and atlas images. Then, the acquired contours are projected onto the target image and assigned the same labels as those on the contoured atlas image. Finally, the contours contributed by each of the selected atlases are merged by label fusion methods on the target image.

This study conducted a retrospective analysis on CT data of 100 female pelvic region patients who underwent radiotherapy treatment at Zhejiang Cancer Hospital from August 24, 2022 to May 31, 2023. The data used in this study was approved by the Medical Ethics Committee of Zhejiang Cancer Hospital (IRB-2023-175). The ethics committee waived the requirement for informed consent. These data were obtained from the hospital database on June 5, 2023 for research purposes. To minimize data degradation from imaging artifacts, patients who had received contrast agents or metal implants (e.g., hip prosthetics) were exempt from enrollment. This collection of images was then divided into two subsets: 70 images were selected randomly to construct the atlas dataset and 30 images out of the atlas dataset were assigned as the validation set. Five contours were manually delineated by an expert radiation oncologist including the bladder, rectum, bone marrow as well as left and right femoral heads. The CT images were acquired using a GE LightSpeed CT scanner (General Electric Healthcare; Milwaukee, WI) and a Brilliance CT Big Bore scanner (Philips Medical Systems, Cleveland, OH, USA). To the best of our knowledge, there is no evidence indicating that different

CT scanners have a significant impact on atlas segmentation results. Therefore, we did not follow any extra criteria when selecting images.

Segmentation was then performed using Raystation v9.0 (RaySearch Laboratories AB, Stockholm, Sweden) based on a DIR algorithm along with rigid registration before segmentation. A hybrid registration (Anatomically Constrained Deformation Algorithm, ANACONDA) combines image similarity with anatomical information [24] as provided by contoured image sets was applied. The five highest performing atlases were integrated into the collaborative registration contour in the label fusion step. Majority voting was then used to assign a label to each voxel that most segmentation agree on for five candidate atlas images. Despite the recognition of potential improvements via the increase of the atlas, we maintained the size at five to balance quality gains against computational expenses.

## Subset atlas grouping approach (MAS-SAGA)

This study proposed a subgrouping method for atlas selection to better identify the best-fitting atlas. An overall framework of the MAS-SAGA for pelvic segmentation was illustrated in Fig 1. Firstly, atlases were grouped according to their volume features in each contour. Four distinct groups were derived following the application of K-means clustering. Subsequently, the target image was assigned to specific subgroups after estimating contour volumes.

**Volume feature extraction.** The goal of this study is to investigate whether volume feature metric atlas selection is better than the intensity-based similarity metric approach. We employed a subset atlas grouping approach based on volume features. For each atlas Ai, we calculated the volume of each contour, $V(A_i) = (V_1, ..., V_n)$. Contours including the body external, bladder, rectum, femoral head, and bone marrow were obtained. Due to the variety in organ volume that may cause discrepancies when registering larger structures to smaller ones, a preliminary step was taken to normalize the volumes within the bounds of [0,1] and minimize negative impacts on segmentation accuracy.

**Feature clustering.** In MAS application, poor contour propagation is usually caused by large volume deformation between the atlas image and target image. Previous studies have suggested that reducing the relative volume ratio can lead to better segmentation results [25]. The subgrouping method is a technique within atlas frameworks. For example, it can be applied in the segmentation of tumor targets at various stages. Employing grouping methods

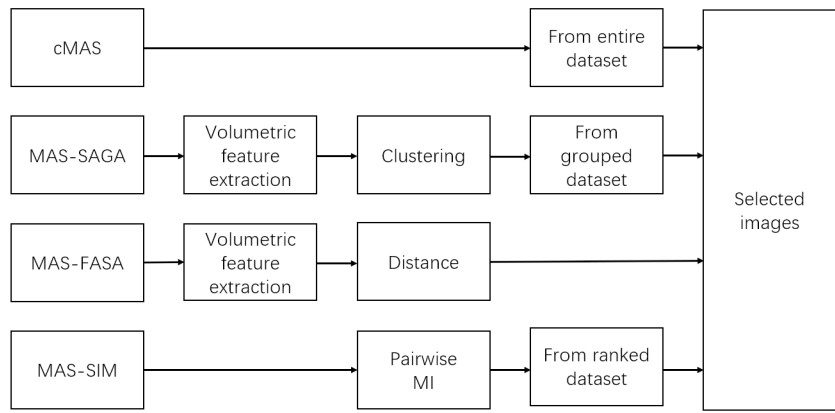

**Fig 1. Schematic of the atlas selection method.**

decreases the likelihood of uncertainties in deformation algorithms. In this research, we utilize Gaussian Mixture Models (GMM) to create subset atlases that reflect volume variations within a target dataset to improve segmentation accuracy.

For each image Ai in atlas dataset, the feature vector $\vec{V}_{atlas}(A_i)$ is formed of normalized volume values as described above. The dimension of $\vec{V}_{atlas}(A_i)$ is equal to the number of contour labels in each image.

A major contribution of this work is to divide the atlas into subsets by volume feature clustering. Given a set of n-dimensional vector $\vec{V}_{atlas}(A_i), i = \{1,\ldots,n\}$, k-means clustering aims to partition the N atlas images into k ($\leq$ N) sets

$$\arg\min_C \sum_{i=1}^{N}\sum_{x\in C_i}\left|V - \mu_i\right|^2$$

where μi is the center of i-th clustering. The atlas dataset A then was divided into k subgroups here $A = \bigcup_{j=1}^{k} A_j$.

The GMM divides a set of n images into k clusters so that each image belongs to the cluster whose centroid is closest to it [12]. In this research, four clusters were assigned without compromising segmentation performance or introducing difficulty in cluster selection. As a result, each cluster contains an average of approximately 18 images making the selection process time-saving. The volume of the ROI needs to be estimated prior to the segmentation process, and then the corresponding subgroup is assigned for segmentation.

## Feature-based atlas selection approach (MAS-FASA)

MAS-SAGA defined subsets utilizing volume features and selected by the cMAS method according to similarity metric. Therefore, the image selected for label fusion is not completely selected according to volume features. Furthermore, identifying the atlas selection for the final label fusion process within a commercialized treatment planning system remains challenging. To further clarify the advantages of volume features metric relative to intensity-based similarity metric in selecting atlas images, we proposed a feature-based atlas selection approach (MAS-FASA) which selected corresponding atlas images to combine in a label fusion process directly.

The distance of volume feature D is used to predict the registration performance of $A_i$ image in the Euclidean dataset and target images.

$$D = \left|\vec{V}_{atlas}(A_i) - \vec{V}_{target}\right|, i = \{1,\ldots,N\}$$

where $\vec{V}_{atlas}(A_i)$ and $\vec{V}_{target}$ denote the volumetric feature vectors of the atlas image and target image, respectively. The identical scaling transformation was applied to the feature vector of the target image. Atlas images with the lowest D were selected with the same number in cMAS and MAS-SAGA. This approach expedites the search for the most fitting image, achieved by computing the distance between $\left(\vec{V}_{atlas}(A_i), \vec{V}_{target}\right)$ without demanding image-to-image similarity evaluations. In this study, MAS-FASA was used to search five atlas images with the minimized distance of volume feature D for each target image. Then thirty independent atlas subsets were created corresponding to each target image.

## Similarity selection (MAS-SIM)

To establish an intensity similarity metric atlas searching method that can be compared with MAS-FASA, MAS-SIM was proposed to further verify the effectiveness of the volume feature metric method. Since acquiring exact atlases for cMAS remains challenging as mentioned in

section 3, a routine to the Insight Segmentation and Registration Toolkit (ITK) was implemented to calculate the mutual information similarity between target and atlas images. These routines compute the MI between two images after rigid registration using the method of Mattes [26]. Mutual information (MI) is a common metric that provides a measure of the intensity similarity between two images due to its robustness to outliers and calculation efficiency [27]. MI assesses the image registration performance and will be maximized when two images are most accurately registered. To identify corresponding images for target volumes within this study, we computed the pairwise MI between every target image and candidate atlas before ranking the intensity similarity of atlas images and finding out the potential matching images to be selected. Then the same size of atlas images with MAS-FASA which were selected with intensity similarity metric were also constructed followed by the same label fusion approaches mentioned in section 3.

## Data analysis

The Dice similarity coefficient (DSC) was used to evaluate the propagation performance of each contour. $DSC(A, B) = 2|A \cap B| / (|A| + |B|)$, Where A was the segmented contour and B was the ground truth contour. DSC obtained by MAS was compared to that one obtained by MAS-SAGA and MAS-FASA. The 95% Hausdorff distance (95HD) was also used which represents the largest surface-to-surface separation among the closest 95% of surface points. The difference between the MAS-SAGA/MAS-FASA approaches and the MAS approach was tested for statistical significance using a two-tailed, paired t-test, assessed by a 0.05 significance level

## Result

### Sub-grouping and auto-segmentation procedure

The 100 atlas samples that were selected are representative, with bladder volumes ranging from 70.89cc to 437.09cc and rectal volumes ranging from 21.3cc to 115.04cc. To evaluate the performance of the proposed atlas selection approach on the MAS segmentation task, we addressed five contours using cMAS, MAS-SAGA, MAS-FASA, and MAS-SIM. In this work, we employed a cross-validation strategy, randomly assigning 30 images as the test set, while the remaining images were assigned to the atlas dataset. The atlas set was randomly selected three times with the following group results: (21, 19, 17, and 13)(20, 14, 15, and 21), and (17, 16, 18, and 19). The cMAS approach used a comprehensive atlas dataset comprising 70 images. Meanwhile, MAS-SAGA constructed four subgroups with variable sizes to generate the best matching images. For MAS-FASA or MAS-SIM, five best-matching atlases were directly selected based on feature distance or similarity from a dataset of 70 atlas images. These selected atlases were then used in the label fusion process to generate the final segmentation results.

### Segmentation performance

Table 1 presents the Dice Similarity Coefficient (DSC) and 95th Percentile Hausdorff Distance (95HD) of cMAS, MAS-SAGA, MAS-FASA, and MAS-SIM for five contours on 30 pelvic cancer patients. Compared to cMAS, the sub-grouping method MAS-SAGA has improved the segmentation accuracy. For the bladder, the Dice Similarity Coefficient (DSC) values were ($0.83 \pm 0.09$) compared to ($0.69 \pm 0.15$), and for the rectum, the values were ($0.70 \pm 0.07$) compared to ($0.56 \pm 0.16$), with statistically significant results. The feature-based method MAS-FASA, in comparison to the similarity-based method MAS-SIM, also achieved good performance. The DSC values for the bladder and rectum were ($0.79 \pm 0.04$) and ($0.67 \pm 0.09$) respectively, slightly lower than those of MAS-SAGA. The 95th percentile Hausdorff Distance

**Table 1. Mean and standard deviation of the average dice score and 95% Hausdorff distance (in cm) incurred by the different method of atlas selection strategies.**

|          |       | Bladder | Rectum | Left Femoral Head | Right Femoral Head | Bone |
|----------|-------|---------|--------|-------------------|--------------------|------|
| cMAS     | DSC   | 0.69 ± 0.15 | 0.56 ± 0.16 | **0.92 ± 0.05** | 0.91 ± 0.04 | **0.92 ± 0.03** |
| MAS-SAGA | DSC   | **0.83 ± 0.09** | **0.70 ± 0.07** | 0.91 ± 0.04 | 0.91 ± 0.02 | 0.91 ± 0.06 |
| MAS-SIM  | DSC   | 0.67 ± 0.13 | 0.49 ± 0.18 | 0.89 ± 0.04 | 0.88 ± 0.03 | 0.89 ± 0.06 |
| MAS-FASA | DSC   | 0.79 ± 0.04 | 0.67 ± 0.09 | 0.90 ± 0.05 | **0.92 ± 0.05** | 0.91 ± 0.04 |
| cMAS     | 95HD  | 1.77 ± 0.34 | 1.01 ± 0.38 | 0.49 ± 0.40 | 0.56 ± 0.16 | 1.01 ± 0.38 |
| MAS-SAGA | 95HD  | 1.38 ± 0.20 | **0.78 ± 0.21** | **0.35 ± 0.13** | **0.41 ± 0.06** | **0.71 ± 0.07** |
| MAS-SIM  | 95HD  | 1.83 ± 0.31 | 1.13 ± 0.33 | 0.52 ± 0.32 | 0.53 ± 0.12 | 0.99 ± 0.18 |
| MAS-FASA | 95HD  | **1.35 ± 0.32** | 1.13 ± 0.28 | 0.37 ± 0.14 | 0.43 ± 0.06 | 0.84 ± 0.13 |

(95HD) values for each method showed a similar trend, with the sub-grouping method MAS-SAGA achieving the best results for the rectum (0.78 ± 0.21 cm), left femoral head (0.35 ± 0.13 cm), right femoral head (0.41 ± 0.06 cm), and bone (0.71 ± 0.07 cm). It is worth noting that the two similarity-related atlas selection methods, cMAS and MAS-SIM, obtained similar results and showed no statistically significant differences are shown in Fig 2 and 3.

## Atlas ranking of strategies

The comparison between MAS-FASA and MAS-SIM regarding the selection of the best-fitting atlases is depicted in Fig 4, illustrating the outcomes of atlas selection derived from a randomly assigned grouping. In the analysis involving the separate application of both methods to the set of 30 test images, five best-fitting atlases were identified for each image, resulting in a total of 300 candidate atlases. It was discerned that the two approaches coincided in selecting the same best-fitting image on only 12 occasions (4.0%). Among this array of atlases, image #41 emerged as the most frequently chosen by both approaches, having been selected in 18 instances, with two instances of concurrent selection by both methods. However, in the majority of cases, the two approaches diverged in their preferences for atlas selection. Within the entire set of 70 atlas images, a subset of 3 (4.3%) atlas images remained unselected (image #22, image #48, image #51)

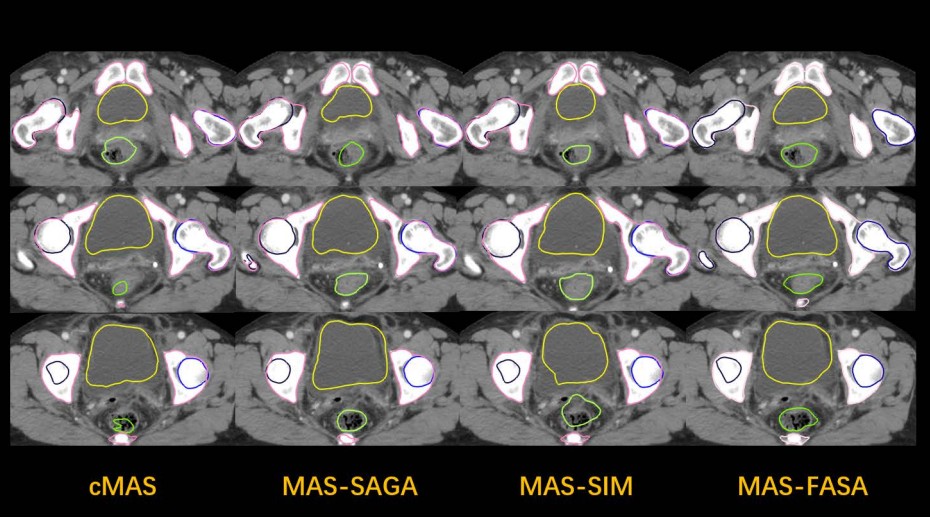

**Fig 2. This figure represents the segmentation performance for cMAS, MAS-SAGA, MAS-SIM, and MAS-FASA.**

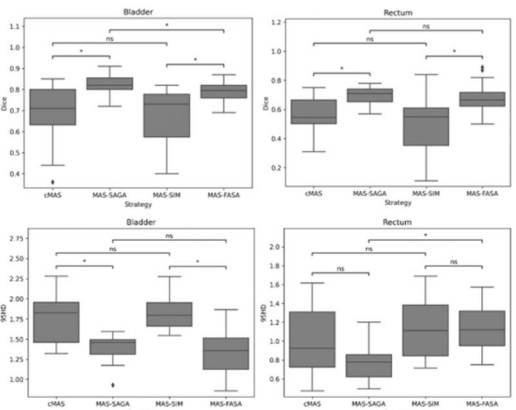

**Fig 3. Dice and 95HD results of the different selection strategies.**

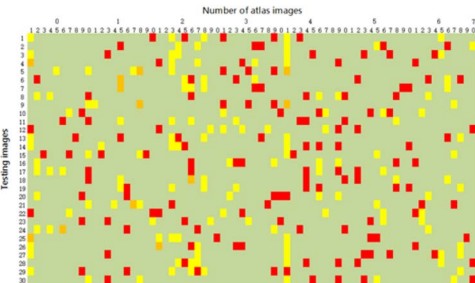

**Fig 4. The images selected in the MAS-SIM and MAS-FASA approaches contain 30 \* 70 color patches. The number of rows represents 30 testing images and the number of columns represents 70 atlas images. Each row contains 5 images selected by MAS-SIM indicated in yellow, and 5 images selected by MAS-FASA indicated in red. The orange patches indicate those images were selected by both MAS-SIM and MAS-FASA. The green patches indicate that the image was not selected.**

## Computation time

The average time cost of the four methods were shown in Table 2. The offline time was the time for subgrouping or calculating the similarity of image pairs. The online time was the best-fitting image selected from the atlas dataset and propagated to the target image. For the MAS-SAGA, clustering takes approximately 2 minutes, and only one approach was required for the entire dataset. MAS-FASA required the contour volume statistics of the images in the dataset but did not increase the preparation time. MAS-SIM required the longest offline time due to the similarity calculation for each image pairwise. On a computer with Intel(R) Xeon(R) Gold 6132 CPU @ 2.60GHz 2.59 GHz (2 processors) and 128GB memory, the time to calculate a pairwise were about 50s. It took about 58.3 minutes for a dataset of 70 images.

## Discussion

The conventional MAS has its limitation on large volume registration which may cause unexpected propagation results in the second step mentioned above. In this paper, we introduced two novel atlas selection methods, MAS-SAGA and MAS-FASA, for multi-atlas-based segmentation (MAS) in medical imaging. We first proposed a MAS-SAGA method that combines

**Table 2. Average time cost of cMAS, MAS-SAGA, MAS-SIM, and MAS-FASA.**

|              | cMAS | MAS-SAGA | MAS-SIM | MAS-FASA |
|--------------|------|----------|---------|----------|
| Offline(min) | N/A  | 2        | 58.3    | N/A      |
| Online(min)  | 6.3  | 2.6      | 0.70    | 0.69     |

Although MAS-FASA performed slightly worse than MAS-SAGA, it is considerably faster, with a mean of 41.6 seconds, as only one step of registration needs to be performed online. The cMAS approach register all atlas images which had slightly better results as reported previously but the computation time was inevitably large (6.3min). MAS-SAGA reduces the computation time significantly (2.6min) by sub-grouping atlases.

the subset atlas grouping approach with volumetric selection and employs a k-means clustering algorithm to partition the atlases based on their features. Experimental evaluation indicates that MAS-SAGA outperforms conventional cMAS by integrating both volumetric and similarity information from atlases. Moreover, our second method, MAS-FASA, selects the most suitable atlases based on their voxel-wise volumetric similarity with the target image, thereby reducing computational time without degrading accuracy. Quantitative analysis shows that both MAS-SAGA and MAS-FASA significantly improve computational efficiency while maintaining comparable segmentation quality to cMAS.

Previous studies evaluating the performance of multiple atlas-based segmentation approaches have yielded varying results depending on the specific algorithm utilized. Different warping techniques can affect the fusion. Some algorithms achieve deformation by controlling the displacement of feature points, while others use finite element methods. Different algorithms and parameter settings handle various types of organs and boundaries differently, leading to varying final results. One such investigation focused on five atlas-based segmentation tools used to delineate the prostate and surrounding structures, including the bladder and rectum [28]. Findings indicated unsatisfactory prostate contouring for the bladder (mean DSC = $0.59 \pm 0.15$ cm)(mean 95HD = $2.85 \pm 1.31$ cm) and rectum (mean DSC = $0.49 \pm 0.12$ cm, 95HD = $1.65 \pm 0.37$ cm). However, our current research, employing cMAS with Raystation software, achieved comparable outcomes relative to the earlier report for the bladder (mean DSC = $0.69 \pm 0.15$, mean 95HD = $1.77 \pm 0.34$ cm) and rectum (mean DSC = $0.56 \pm 0.16$, mean 95HD = $1.01 \pm 0.38$ cm), though they still presented room for improvement. Our novel approach, MAS-SAGA, further improved the precision of these segmentations for the bladder (mean DSC = $0.83 \pm 0.09$, mean 95HD = $1.38 \pm 0.20$ cm) and rectum (mean DSC = $0.70 \pm 0.07$, mean 95HD = $0.78 \pm 0.21$ cm) as compared to traditional cMAS using similarity selection strategies.

The size of atlas datasets can greatly affect both the accuracy and time efficiency of segmentation implementation. Although large datasets have the potential to produce high-quality segmentations, their increased processing time could become inefficient. According to ANACONDA's recommendations, a smaller atlas comprising just 25 images would suffice, with subsequent evaluation beginning from the fifth image. Some previous works investigate how many images are adequate when necessary to enhance segmentation outputs. Kim et al evaluated three different atlas libraries for ABAS in groups of 20, 40, or 60 [29]. They found a poor performance of bladder segmentation with a DSC < 0.6 and a mean HD > 40 mm in all libraries. Increasing the size of libraries did not improve the results of segmentation. Our clustering procedure allowed us to extract a representative set of atlas images from the candidate images pool, each subgroup consisted of fewer than 25 images, resulting in substantially enhanced segmentation performance compared to conventional methods.

We believe that the success of segmentation depends on whether the task falls within the reasonable computational scale of the deformation registration algorithm. The results showed that SAGA achieved better performance because it initially filters out large-volume deformations, which pose significant challenges to deformations. Intensity-based metrics are the most

frequently applied atlas selection method for medical image registration. Although the difficulty of deforming dissimilar images is much greater than that of deforming similar images, ROI volume deformation may be quite the opposite. Images with high overall similarity but significant differences in ROI often result in poor segmentation. However, studies have shown that differences in overall image similarity may not accurately represent variations in local regions of interest. Few publications have examined the impact of employing both similarity and volumetric features for atlas selection on segmentation outcomes. Our study was the first exploration revealing dissimilar performance from intensity versus volume-driven selection techniques, with merely 4% of chosen atlases being consistent between them. Notably, the proposed volumetric feature-based selection led to superior segmentation accuracy of the bladder and rectum in the MAS-SAGA and MAS-FASA, coupled with accelerated computation times beneficial for clinical implementation.

One limitation of current research lies in its dependency on selecting the external contour as the primary region of interest for deformation registration-based segmentation. This choice may lead to variations in segmentation quality due to variable slice thicknesses, patient body size, and bladder volumes [8]. Therefore, optimized scanning protocols and image pre-processing would probably enhance the quality of segmentation. Additionally, more accurate segmentation performances could be acquired by determining the volume of ROIs effectively during the examination. Further investigations should concentrate on developing effective methods to estimate the volume of the ROI. Such methods might involve computing the bladder's dimensions and quantifying the intersection area of the rectum in imaging slices. Another challenge arises from deformable registration, wherein the regularization terms of different deformable registration methods vary, leading to different degrees of algorithmic flexibility. This study achieved relatively favorable results by employing the ANACONDA registration method in RayStation. Additionally, the required number of groups needs to be determined based on the specific deformable algorithm when implementing other deformable registration methods.

## Conclusion

This study proposed a subgrouping method for MAS, which aided in searching for the most fitting atlases based on multiple features. The proposed MAS-SAGA revealed improvements in segmentation performance compared to conventional MAS (cMAS) approach, with the DSC for the bladder and rectum escalating from $0.69 \pm 0.15$ to $0.83 \pm 0.09$ and from $0.56 \pm 0.16$ to $0.70 \pm 0.07$, respectively. Furthermore, we conducted a comparative analysis between two atlas selection methods, similarity and volume features, and found that the consistency in candidate atlas selection between the two methods was only 4%, indicating significant disagreement between the two approaches. Hence, the integration of volume features into atlas search contributes to enhancing the segmentation performance of MAS.

## Acknowledgments

The authors would like to acknowledge research assistant, Yizhen Wang, who has shown a strong interest in image processing technics and actively participated in this research. His assistance in conducting software testing on the data and his contributions to data visualization are greatly acknowledged.

## Author contributions

**Conceptualization:** Guoping Shan, Xue Bai, Yun Ge.

**Writing – original draft:** Binbing Wang.

**Writing – review & editing:** Binbing Wang.

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
