## [Decision Letter · Decision Letter 0]

26 Mar 2024

PONE-D-23-24775
A feature-based approach for atlas selection in automatic pelvic segmentation
PLOS ONE

Dear Dr. Wang,

Thank you for submitting your manuscript to PLOS ONE. After careful consideration, we feel that it has merit but does not fully meet PLOS ONE’s publication criteria as it currently stands. Therefore, we invite you to submit a revised version of the manuscript that addresses the points raised during the review process.
 
Specially, please clarify the main contribution of the manuscript.

We look forward to receiving your revised manuscript.

Kind regards,

Jose Gerardo Tamez-Peña, PhD

Academic Editor

PLOS ONE

4. PLOS requires an ORCID iD for the corresponding author in Editorial Manager on papers submitted after December 6th, 2016. Please ensure that you have an ORCID iD and that it is validated in Editorial Manager. To do this, go to ‘Update my Information’ (in the upper left-hand corner of the main menu), and click on the Fetch/Validate link next to the ORCID field. This will take you to the ORCID site and allow you to create a new iD or authenticate a pre-existing iD in Editorial Manager. Please see the following video for instructions on linking an ORCID iD to your Editorial Manager account: " xlink:type="simple">https://www.youtube.com/watch?v=_xcclfuvtxQ".

 [This study was financially supported by the Medical Science and Technology Project of 

Zhejiang Province (2021PY039), the Natural Science Foundation of Zhejiang Province

(LSY19H180002).].  

[This study was financially supported by the Medical Science and Technology Project of Zhejiang Province (2021PY039), the Natural Science Foundation of Zhejiang Province (LSY19H180002). The authors would like to acknowledge research assistant, Wang Yizhen, who has shown a strong interest in image processing technics and actively participated in this research. His assistance in conducting software testing on the data and his contributions to data visualization are greatly acknowledged.]

 [This study was financially supported by the Medical Science and Technology Project of 

Zhejiang Province (2021PY039), the Natural Science Foundation of Zhejiang Province

(LSY19H180002).]. 

7. Please provide a complete Data Availability Statement in the submission form, ensuring you include all necessary access information or a reason for why you are unable to make your data freely accessible. If your research concerns only data provided within your submission, please write "All data are in the manuscript and/or supporting information files" as your Data Availability Statement.

8. We note that Figure 2 in your submission contain copyrighted images. All PLOS content is published under the Creative Commons Attribution License (CC BY 4.0), which means that the manuscript, images, and Supporting Information files will be freely available online, and any third party is permitted to access, download, copy, distribute, and use these materials in any way, even commercially, with proper attribution. For more information, see our copyright guidelines: http://journals.plos.org/plosone/s/licenses-and-copyright.

1. You may seek permission from the original copyright holder of Figure 2 to publish the content specifically under the CC BY 4.0 license. 

Reviewers' comments:

Reviewer's Responses to Questions

**Comments to the Author**

1. Is the manuscript technically sound, and do the data support the conclusions?

Reviewer #1: Yes

Reviewer #2: Yes

2. Has the statistical analysis been performed appropriately and rigorously? 

Reviewer #1: Yes

Reviewer #2: Yes

3. Have the authors made all data underlying the findings in their manuscript fully available?

Reviewer #1: Yes

Reviewer #2: Yes

4. Is the manuscript presented in an intelligible fashion and written in standard English?

Reviewer #1: Yes

Reviewer #2: Yes

5. Review Comments to the Author

Reviewer #1: In this paper, the authors propose an atlas selection procedure (subset atlas grouping approach, MAS-SAGA) that utilises both image similarity and volume features for selecting the best-fitting atlases for contour propagation. The authors did good work and were interested in the readers. The following review comments are recommended, and the authors are invited to explain and modify.

1 The main contributions of the manuscript are not clear. The main contributions of the &article must be very clear, and it would be better to summarise them into 3-4 points at the end of the introduction. &

2 What is the logic behind utilising both image similarity and volume features for selecting the best-fitting atlases for contour propagation?

3 When writing phrases like “Medical image segmentation predefines normal tissues for the purpose of their protection in radiation therapy planning, thus having broad applications in the field of radiation therapy," it should cite related works in order to sustain the statement: 10.1155/2022/2665283; 10.1155/2023/2345835.

4 This study proposed a subgrouping method for atlas selection to better identify the best-fitting atlas, why did it choose the subgrouping method?

5 Why did the authors not apply the deep learning approach to automatic pelvic segmentation?

6 The authors should mention the implementation challenges.

7 Moreover, it should be noticed that the clinical appliance has to be decided by medical professionals since the existing differences between the real image and the one generated by the proposed model could be substantial in the medical field.

Reviewer #2: Dataset is big enough to offer statistically relevant results.

Dataset was separated in good proportions between a construction subset and a testing subset.

Commonly used algorithms were involved in this study to assess the data.

According to the author,

Data cannot be shared publicly because it contains personal information restricted to

use. Data are available from the Zhejiang Cancer Hospital Institutional Data Access /

Ethics Committee for researchers who meet the criteria for access to confidential data.

Data cannot be shared publicly because it contains personal information restricted to

use. Data are available from the Zhejiang Cancer Hospital Institutional Data Access /

Ethics Committee for researchers who meet the criteria for access to confidential data.

Beside minor points to correct, the manuscript is very well written ad understandable.

6. PLOS authors have the option to publish the peer review history of their article (what does this mean?). If published, this will include your full peer review and any attached files.

Reviewer #1: No

Reviewer #2: **Yes: **Johann Hêches

---

## [Author Response · Author response to Decision Letter 0]

18 Apr 2024

response to Reviewer #1

1 The main contributions of the manuscript are not clear. The main contributions of the

‎article must be very clear, and it would be better to summarise them into 3-4 points at the end of the introduction. ‎

Thank you for your suggestions. We have incorporated the following description into the introduction:

 A subgrouping atlas search approach was proposed, wherein atlases are distributed based on volume features within the atlas database. This enables the search strategies to select the most fitting atlases considering both similarity and volume features, thereby enhancing segmentation accuracy.

 To further clarify the advantage of volume features in selecting atlases, this study then ranked the most fitting atlases obtained from two atlas selection approaches, based on similarity and volume features, and compared their differences in priority when selecting candidate atlases.

 A comparison of the execution time efficiency of the four proposed atlas search methods was also performed.

2 What is the logic behind utilising both image similarity and volume features for selecting the best-fitting atlases for contour propagation?

We have included the following content in the discussion section: “We believe that the success of segmentation depends on whether the task falls within the reasonable computational scale of the deformation registration algorithm. The results showed that SAGA achieved better performance because it initially filters out large-volume deformations, which pose significant challenges to deformations.” “Although the difficulty of deforming dissimilar images is much greater than that of deforming similar images, ROI volume deformation may be quite the opposite. Images with high overall similarity but significant differences in ROI often result in poor segmentation.”

3 When writing phrases like “Medical image segmentation predefines normal tissues for the purpose of their protection in radiation therapy planning, thus having broad applications in the field of radiation therapy," it should cite related works in order to sustain the statement: 10.1155/2022/2665283;.

We have added the corresponding literature

4 This study proposed a subgrouping method for atlas selection to better identify the best-fitting atlas, why did it choose the subgrouping method?

We have included the following content in the method section:” The subgrouping method is a technique within atlas frameworks. For example, it can be applied in the segmentation of tumor targets at various stages. Employing grouping methods decreases the likelihood of uncertainties in deformation algorithms.”

Previous grouping methods relied on subjective criteria. The innovation of this study lies in proposing a volume feature-based grouping method using GMM clustering.

5 Why did the authors not apply the deep learning approach to automatic pelvic segmentation?

Considering the significant impact of deep learning in the field of image segmentation, we have added the following description to the introduction：

Recently, MAS has been challenged by deep learning-based segmentation. Deep learning based segmentation employs deep neural network models to learn features and semantic information from images, achieving excellent segmentation results in medical image segmentation[1]. However, this method also exhibits certain drawbacks, such as challenges in data acquisition and annotation, limited model generalization capability, and poor interpretability. When employed in new tasks or with diverse types of imaging data, their performance may significantly decline[2]. In contrast, MAS is an interpretable method and does not require a large number of annotated images for training. Therefore, it is still being used in some fields, such as brain segmentation[3] and dose accumulation assessment in radiotherapy[4, 5] and has not been replaced by deep learning. however, the lack of precise feature classification during the atlas search and deformation processes in MAS leads to a segmentation accuracy inferior to that achieved by deep learning methodologies.

6 The authors should mention the implementation challenges.

To clarify the challenges encountered in the implementation, we have added the following content to the discussion section:

Another challenge arises from deformable registration, wherein the regularization terms of different deformable registration methods vary, leading to different degrees of algorithmic flexibility. This study achieved relatively favorable results by employing the ANACONDA registration method in RayStation. Additionally, the required number of groups needs to be determined based on the specific deformable algorithm when implementing other deformable registration methods.

7 Moreover, it should be noticed that the clinical appliance has to be decided by medical professionals since the existing differences between the real image and the one generated by the proposed model could be substantial in the medical field.

As pointed out by the reviewers, the final results require validation by clinical experts. Similar to most "automatic" segmentation methods, our findings act as a guide for clinical experts and are not entirely automated. Modifications should be made before implementation. While many articles have discussed modification time and complexity, subjective evaluations were not conducted in this study.

The clinical acceptance of a model depends on subjective evaluations as well as factors such as segmentation time and the robustness of implementation. This study concentrates on describing the grouping strategy of MAS and objectively testing DSC and HD, indicating the clinical applicability of this model.

response to Reviewer #2

Manuscript #PONE-D-23-24775

“A feature-based approach for atlas selection in automatic pelvic segmentation”

Peer-reviewed by Dr. Johann Hêches

Overall, the proposed manuscript is well written and of interest.

Any research that leads to improving accuracy and computational cost of automatic image segmentation methods is very welcomed as manual segmentation is exhausting.

Please find my comments underneath, may they hopefully help you to improve the quality of this manuscript.

A/

“These routines compute the MI between two images after rigid registration using the method of Mattes[19]. Mutual information (MI)”

Consider defining the “mutual information (MI)” abbreviation at its first occurrence in the text.

Thank you for pointing out the shortcomings of the article. The term 'Mutual information' has been abbreviated and defined upon its first occurrence. These similarity metric includes similarity index[6], the sum of squared difference of image intensity[7], correlation coefficient [8], and mutual information (MI) [9].”

B/

“(, ) = 2| ∩ |/|| + ||”

A parenthesis is missing in this formula, |B| should be part of the fraction denominator.

Thank you for identifying the flaws in the article. They have been addressed as: DSC(A,B)=2|A∩B|/(|A|+|B|)

C/

“We also suggested a feature-based atlas selection approach (MAS-FASA) that atlas with the closest feature space distance is selected as the candidate for MAS segmentation without including any additional atlases, thus reducing the atlas search time”

Either some punctuation might be missing or the sentence should be rewritten. The sentence is understandable but it was very confusing to read at first.

Thank you for pointing out the flaws in the article. The following modifications have been made:

We also suggested a feature-based atlas selection approach (MAS-FASA), where the atlas with the closest feature space distance is selected as the candidate for MAS segmentation. This method eliminates the need for additional atlases, resulting in reduced atlas search time.

D/

Conclusion is missing for the manuscript ?

Thank you for pointing out the shortcomings of the article. A conclusion section has been added with the following content:

Conclusion

This study proposed a subgrouping method for MAS, which aided in searching for the most fitting atlases based on multiple features. The proposed MAS-SAGA revealed improvements in segmentation performance compared to conventional MAS (cMAS) approach, with the DSC for the bladder and rectum escalating from 0.69±0.15 to 0.83±0.09 and from 0.56±0.16 to 0.70±0.07, respectively. Furthermore, we conducted a comparative analysis between two atlas selection methods, similarity and volume features, and found that the consistency in candidate atlas selection between the two methods was only 4%, indicating significant disagreement between the two approaches. Hence, the integration of volume features into atlas search contributes to enhancing the segmentation performance of MAS.

---

## [Decision Letter · Decision Letter 1]

28 May 2024

PONE-D-23-24775R1
A feature-based approach for atlas selection in automatic pelvic segmentation
PLOS ONE

Dear Dr. Wang,

Thank you for submitting your manuscript to PLOS ONE. After careful consideration, we feel that it has merit but does not fully meet PLOS ONE’s publication criteria as it currently stands. Therefore, we invite you to submit a revised version of the manuscript that addresses the points raised during the review process.

We look forward to receiving your revised manuscript.

Kind regards,

Jose Gerardo Tamez-Peña, PhD

Academic Editor

PLOS ONE

Journal Requirements:

Reviewers' comments:

Reviewer's Responses to Questions

**Comments to the Author**

1. If the authors have adequately addressed your comments raised in a previous round of review and you feel that this manuscript is now acceptable for publication, you may indicate that here to bypass the “Comments to the Author” section, enter your conflict of interest statement in the “Confidential to Editor” section, and submit your "Accept" recommendation.

Reviewer #2: All comments have been addressed

Reviewer #3: All comments have been addressed

2. Is the manuscript technically sound, and do the data support the conclusions?

Reviewer #2: Yes

Reviewer #3: Partly

3. Has the statistical analysis been performed appropriately and rigorously? 

Reviewer #2: Yes

Reviewer #3: Yes

4. Have the authors made all data underlying the findings in their manuscript fully available?

Reviewer #2: Yes

Reviewer #3: Yes

5. Is the manuscript presented in an intelligible fashion and written in standard English?

Reviewer #2: Yes

Reviewer #3: Yes

6. Review Comments to the Author

Reviewer #2: All my comments, as well as the comments from the other reviewer, appears to be fully addressed. All good on my side.

Reviewer #3: Overall this manuscript about multi atlas segmentation using subset atlas group approach is well-written. The stated aim of comparing cMAS with new technique is clear. More guidance should be provided to the readers to make the results more accessible.

The idea is good; however, I have some comments about improving the current version.

In the Introduction section, the advantage of MAS over DL has been discussed multiple times.

What is the rationale behind choosing image similarity and volumetric features?

Can you explain if the atlases selected are the representative sample of population? (That represents better anatomical variability?)

Do you think different warping techniques and parameters would affect the fusion?

Can you explain the robustness of the technique for example, atlas with different scanner characteristics and sample demographics?

Please explain the cross validation results (training and test set)?

7. PLOS authors have the option to publish the peer review history of their article (what does this mean?). If published, this will include your full peer review and any attached files.

Reviewer #2: **Yes: **Johann Hêches

Reviewer #3: No

---

## [Author Response · Author response to Decision Letter 1]

11 Jul 2024

Overall this manuscript about multi atlas segmentation using subset atlas group approach is well-written. The stated aim of comparing cMAS with new technique is clear. More guidance should be provided to the readers to make the results more accessible.

The idea is good; however, I have some comments about improving the current version.

In the Introduction section, the advantage of MAS over DL has been discussed multiple times. What is the rationale behind choosing image similarity and volumetric features?

Thank you for your suggestions. We have included the following content in the introduction section: Our study is based on the following hypothesis that similarity-based atlas selection methods tend to search for atlases with high overall image similarity. However, when dealing with large-scale deformation, the accuracy of deformation modeling is often compromised due to the involvement of significant nonlinear deformations, geometric complexities, and the impact of intricate boundary conditions and constraints. It may lead to instability in segmentation results, making further refinement challenging.

Can you explain if the atlases selected are the representative sample of population? (That represents better anatomical variability?)

Thank you for your suggestions. We have included the following content in the Result section: The 100 atlas samples that were selected are representative, with bladder volumes ranging from 70.89cc to 437.09cc and rectal volumes ranging from 21.3cc to 115.04cc.

Their volume distributions are shown in the figure. The data were randomly selected from the patient database, and the volume distribution is typical. We believe this approach enhances the robustness of the model.

Do you think different warping techniques and parameters would affect the fusion?

We believe that different warping techniques can affect the fusion. By setting specific Young's Modulus values for each type of organ, the resulting DVF can be adjusted. Some algorithms achieve deformation by controlling the displacement of feature points, while others use finite element methods. Different algorithms and parameter settings vary in handling different types of organs and boundaries, leading to different final results.

According to the reviewers' comments, we have made the corresponding revisions in the Discussion section.

Can you explain the robustness of the technique for example, atlas with different scanner characteristics and sample demographics?

The data we used came from two different types of scanners, the GE LightSpeed CT scanner and the Brilliance CT Big Bore scanner. We did not observe any influence of different scanners on deformation results, nor did we find any relevant reports. It is clear that the demographics of the samples can affect deformation results, so we also considered their representativeness when selecting atlases.

According to the reviewers' comments, we have added the following content to the Method section: To the best of our knowledge, there is no evidence indicating that different CT scanners have a significant impact on atlas segmentation results. Therefore, we did not follow any specific inclusion or exclusion criteria when selecting images.

Please explain the cross validation results (training and test set)?

Thank you for the reviewers' suggestions. Based on these insights, we have supplemented the content in the Results section.

We randomly selected 70 out of 100 images as the atlas set and used the remaining 30 for validation. The training set was randomly selected three times with the following group results: (21, 19, 17, and 13), (20, 14, 15, and 21), and (17, 16, 18, and 19). Subsequently, contour comparison was conducted on the 30 image sets, calculating DSC and 95HD. Unlike deep learning cross-validation methods which evaluate convergence with loss functions on test sets, MAS methods do not exhibit overfitting during training. We tested cMAS, MAS-SAGA, MAS-FASA, and MAS-SIM, with results shown in Table 1. Cross-validation results indicate that incorporating volume features as classification criteria leads MAS-SAGA and MAS-FASA to achieve superior DSC and 95HD scores with lower variance compared to cMAS.

---

## [Editor Report · Decision Letter 2]

6 Jan 2025

A feature-based approach for atlas selection in automatic pelvic segmentation

PONE-D-23-24775R2

Dear Dr. Wang,

We’re pleased to inform you that your manuscript has been judged scientifically suitable for publication and will be formally accepted for publication once it meets all outstanding technical requirements.

Kind regards,

Jose Gerardo Tamez-Peña, PhD

Academic Editor

PLOS ONE
---

## [Editor Report · Acceptance letter]

PONE-D-23-24775R2

PLOS ONE

Dear Dr. Wang,

I'm pleased to inform you that your manuscript has been deemed suitable for publication in PLOS ONE. Congratulations! Your manuscript is now being handed over to our production team.

Kind regards,

on behalf of

Dr. Jose Gerardo Tamez-Peña

Academic Editor

PLOS ONE